

# Foliar nutrient concentrations of six northern hardwood species responded to nitrogen and phosphorus fertilization but did not predict tree growth

Daniel S. Hong[1], Kara E. Gonzales[2], Timothy J. Fahey[3] and Ruth D. Yanai[1]

[1] State University of New York College of Environmental Science and Forestry, Syracuse, NY, United States of America
[2] California Department of Fish and Wildlife, Sacramento, CA, United States of America
[3] Cornell University, Ithaca, NY, United States of America

## ABSTRACT

Foliar chemistry can be useful for diagnosing soil nutrient availability and plant nutrient limitation. In northern hardwood forests, foliar responses to nitrogen (N) addition have been more often studied than phosphorus (P) addition, and the interactive effects of N and P addition have rarely been described. In the White Mountains of central New Hampshire, plots in ten forest stands of three age classes across three sites were treated annually beginning in 2011 with 30 kg N $ha^{-1}$ $y^{-1}$ or 10 kg P $ha^{-1}$ $y^{-1}$ or both or neither–a full factorial design. Green leaves of American beech (*Fagus grandifolia* Ehrh.), pin cherry (*Prunus pensylvanica* L.f.), red maple (*Acer rubrum* L.), sugar maple (*A. saccharum* Marsh.), white birch (*Betula papyrifera* Marsh.), and yellow birch (*B. alleghaniensis* Britton) were sampled pre-treatment and 4–6 years post-treatment in two young stands (last cut between 1988–1990), four mid-aged stands (last cut between 1971–1985) and four mature stands (last cut between 1883–1910). In a factorial analysis of species, stand age class, and nutrient addition, foliar N was 12% higher with N addition ($p < 0.001$) and foliar P was 45% higher with P addition ($p < 0.001$). Notably, P addition reduced foliar N concentration by 3% ($p = 0.05$), and N addition reduced foliar P concentration by 7% ($p = 0.002$). When both nutrients were added together, foliar P was lower than predicted by the main effects of N and P additions ($p = 0.08$ for N × P interaction), presumably because addition of N allowed greater use of P for growth. Foliar nutrients did not differ consistently with stand age class ($p \geq 0.11$), but tree species differed ($p \leq 0.01$), with the pioneer species pin cherry having the highest foliar nutrient concentrations and the greatest responses to nutrient addition. Foliar calcium (Ca) and magnesium (Mg) concentrations, on average, were 10% ($p < 0.001$) and 5% lower ($p = 0.01$), respectively, with N addition, but were not affected by P addition ($p = 0.35$ for Ca and $p = 0.93$ for Mg). Additions of N and P did not affect foliar potassium (K) concentrations ($p = 0.58$ for N addition and $p = 0.88$ for P addition). Pre-treatment foliar N:P ratios were high enough to suggest P limitation, but trees receiving N ($p = 0.01$), not P ($p = 0.64$), had higher radial growth rates from 2011 to 2015. The growth response of trees to N or P addition was not explained by pre-treatment foliar N, P, N:P, Ca, Mg, or K.

Corresponding author
Ruth D. Yanai, rdyanai@syr.edu

## INTRODUCTION

Foliar nutrient concentrations reflect soil fertility, and they can indicate how different species respond to changes in nutrient availability (*Aerts & Chapin III, 2000*). Foliar nutrients also provide a means to evaluate nutrient deficiency and monitor changes due to natural and anthropogenic disturbance (*Vitousek, 1998*; *Hobbie & Gough, 2002*; *Townsend et al., 2007*). The addition of nitrogen (N) or phosphorus (P) can affect foliar concentrations of the other nutrient, either positively (*Güsewell, Koerselman & Verhoeven, 2003*; *Niinemets & Kull, 2005*; *Lu et al., 2013*) or negatively (*Niinemets & Kull, 2005*; *Menge & Field, 2007*; *Ostertag, 2010*). Concentrations of base cations in foliage can also be affected, especially by N addition, which can lead to the mobilization of base cations and soil acidification (*Tian & Niu, 2015*). The consequences can vary; in tropical grasslands, N addition caused higher concentrations of calcium (Ca), magnesium (Mg), and potassium (K) in leaf tissues (*Hamilton et al., 1998*), whereas N addition decreased Ca contents in roots in a semiarid temperate steppe (*Fang et al., 2012*).

Plant responses to changes in nutrient availability vary among species. Most temperate deciduous forests are composed of a mixture of tree species with varying life history and physiological traits, such as nutrient requirements and strategies of nutrient acquisition and conservation. The pioneer tree species, pin cherry (*Prunus pensylvanica* L.f.), exhibits exceptional luxury uptake of N when it is in high supply, remobilizing it when N supply declines during succession (*Marks & Bormann, 1972*). Other tree species in northern hardwood ecosystems are known to differ in cycling and retention of N and P (*Lovett et al., 2004*). For example, sugar maple (*Acer saccharum* Marsh.) causes higher rates of soil N mineralization and nitrification compared to other northern hardwood species (*Lovett et al., 2004*). Similarly, inorganic P availability in soils differs with tree species composition in several eastern deciduous forests (*Boerner & Koslowsky, 1989*).

Foliar N:P has been proposed as an indicator of N *vs.* P limitation of plant growth (*Koerselman & Meuleman, 1996*), but the critical value of this ratio is not universal. Foliar N:P ratios reported to indicate P limitation include 14 in a montane forest in Hawaii (*Herbert & Fownes, 1995*), 15 in a beech forest in Europe (*Ljungström & Nihlgård, 1995*), and 18 in understory vegetation in the Catskill region, NY (*Tessier & Raynal, 2003*). Surprisingly, the N:P ratio distinguishing N from P limitation in northern hardwoods has not been established, perhaps because N has been presumed to be limiting (*Vadeboncoeur, 2010*), and thus P fertilization trials are rare.

Multiple Element Limitation in Northern Hardwood Ecosystems (MELNHE) is the first long-term, full-factorial N by P fertilization study in a temperate forest, with low levels of N and P added to 13 stands in the White Mountains of New Hampshire, USA, beginning in 2011. There have been conflicting reports of nutrient limitation in this forest type, with some studies finding a greater growth response to P and others finding a greater response

to N (*Vadeboncoeur, 2010*). In the MELNHE sites prior to fertilization, fine root growth in nutrient-amended cores suggested P limitation in mid-aged forests and N limitation in mature forests (*Naples & Fisk, 2010*). However, aboveground growth responses after four years of fertilization suggested P limitation in both mid-aged and mature stands, but N limitation in the youngest stands (*Goswami et al., 2018*).

In this paper, we present foliar nutrient concentrations before and after 4–6 years of N and P addition in ten stands in the MELNHE study, focusing on the six most abundant tree species and the elements N, P, Ca, Mg, and K. We report foliar responses to N and P addition and test whether adding one nutrient affects concentrations of other nutrients in foliage. We compare foliar responses to nutrient additions across tree species and stand age classes. Lastly, we assess whether pre-treatment foliar nutrient concentrations, or the N:P ratio, can predict growth response of individual trees to nutrient addition.

## METHODS

### Site description

This study took place in ten stands in the White Mountains of central New Hampshire, USA: six at the Bartlett Experimental Forest (44°03′N, 71°17′W), two at the Hubbard Brook Experimental Forest (43°56′N, 71°44′W), and two at Jeffers Brook (44°02′N, 71°53′W). The stands vary in age and species composition (Table 1). All soils are primarily well drained, acid Spodosols of sandy-loam texture developed in glacial drift deposited approximately 14,000 years ago (*Vadeboncoeur et al., 2014*). The soils vary in fertility because of differences in bedrock and till composition (*Vadeboncoeur et al., 2012*; *Vadeboncoeur et al., 2014*; *Bae et al., 2015*). The climate is humid continental, with annual temperature and precipitation averaging 5.7 °C and 1400 mm at forest headquarters at Hubbard Brook (*Bailey et al., 2003*); small climatic differences exist across the stands reflecting variation in elevation and aspect (Table 1). Wet N deposition in this region decreased from 4–7 kg N ha$^{-1}$ yr$^{-1}$ between 1979 and 2003 to 2–4 kg N ha$^{-1}$ yr$^{-1}$ in 2008 (*NADP Program Office, 2017*).

Species composition in the forest stands is typical of the northern hardwood forest type; we studied the six most dominant tree species across the ten stands. These include two extremely shade-tolerant species, American beech (*Fagus grandifolia* Ehrh.) and sugar maple; two intermediate species, red maple (*A. rubrum* L.) and yellow birch (*Betula alleghaniensis* Britton); and two extremely intolerant species, white birch (*B. papyrifera* Marsh.) and pin cherry. These species include ectomycorrhizal taxa (beech and birches) and arbuscular mycorrhizal taxa (cherry and maples). Species composition varied by stand, mainly due to successional changes; all the mature stands were dominated by sugar maple, American beech and yellow birch, whereas the younger stands included a variable mix of all six species. All the stands regenerated following intensive forest harvest with the age of the stands at the time of sampling ranging from 25 to over 100 years (Table 1).

Experimental plots in these stands were established as a part of a project investigating Multiple Element Limitation in Northern Hardwood Ecosystems (MELNHE) (*Fisk et al., 2014*). Each stand contains four plots that have been treated annually since 2011 with either N (30 kg N/ha/yr as NH$_4$NO$_3$), P (10 kg P/ha/yr as NaH$_2$PO$_4$), or N+P (at the same

**Table 1  Stand descriptions for the Bartlett Experimental Forest (BEF), Hubbard Brook Experimental Forest (HBEF), and Jeffers Brook (JB).**

| Site | Stand | Year cut | Age class | Elevation (m) | Aspect | Slope (%) | Studied species |
|------|-------|----------|-----------|---------------|--------|-----------|-----------------|
| BEF | C1 | 1990 | Young | 570 | SE | 5–20 | American beech, pin cherry, and white birch |
| | C2 | 1988 | Young | 340 | NE | 15–30 | American beech, pin cherry, red maple, white birch, and yellow birch |
| | C4 | 1978 | Mid-Aged | 410 | NE | 20–25 | American beech, pin cherry, red maple, white birch, and yellow birch |
| | C6 | 1975 | Mid-Aged | 460 | NNW | 13–20 | American beech, pin cherry, red maple, white birch, and yellow birch |
| | C8 | 1883 | Mature | 330 | NE | 5–35 | American beech, sugar maple, and yellow birch |
| | C9 | ~1890 | Mature | 440 | NE | 10–35 | American beech, sugar maple, and yellow birch |
| HBEF | HBM | 1970 | Mid-Aged | 500 | S | 10–25 | American beech, red maple, white birch, and yellow birch |
| | HBO | ~1900 | Mature | 500 | S | 25–35 | American beech, sugar maple, and yellow birch |
| JB | JBM | ~1975 | Mid-Aged | 730 | WNW | 25–35 | American beech, pin cherry, sugar maple, white birch, and yellow birch |
| | JBO | ~1900 | Mature | 730 | WNW | 30–40 | American beech, sugar maple, and yellow birch |

rates), and an untreated control. The mass ratio of 3:1 for addition of N and P exceeded the typical foliar ratio in northern hardwood forests (15 to 20) to compensate for occlusion of P in unavailable forms in acid soils (*Wood, Bormann & Voight, 1984*; *Cross & Schlesinger, 1995*). All the plots at Bartlett and those in the mature stands at Hubbard Brook and Jeffers Brook include a 10-m wide buffer surrounding a 30 m × 30 m measurement area. In the mid-aged stands at Hubbard Brook and Jeffers Brook, the measurement areas are 20 m × 20 m with a 5-m buffer.

## Sample collection

Pre-treatment foliage was collected from 2008 to 2010 and post-treatment foliage was collected from 2014 to 2016 (after 4–6 years of treatment) in the last week of July and first week of August (Appendix A); a subset of trees was sampled in more than one pre-treatment year to assess interannual variation. When possible, trees were sampled within the measurement area to avoid edge effects. In 33 cases of 1,356 samples, trees were sampled from the buffer because there were too few trees of a species in the measurement area. Foliar chemistry of these trees was not distinguishable from those sampled in the measurement area ($p$ values ranged from 0.15 to 0.75, depending on the element).

Foliage was collected using a 12-gauge shotgun, except for pre-treatment foliage in the youngest stands at Bartlett, which were collected with a pole pruner as the trees were not very tall. Leaves were collected from sun-exposed portions of the middle to upper canopy. When possible, the same trees were sampled over time. Pre-treatment, across multiple years, the average number of trees of each species in each plot was 5.5 ± 0.2 (SE). Post-treatment, we aimed to sample 3 trees of each species in each plot, but in 17 of 157 plot-species combinations, only one or two individuals of a species were found in the plot, resulting in an average of 2.9 ± 0.04 trees per species per plot.

Tree diameters were measured on all trees ≥10 cm diameter at breast height (DBH) in August of 2011 and 2015 (*Goswami et al., 2018*). Some of the trees that were sampled for

foliar chemistry were not large enough (≥10 cm DBH) to be tracked for diameter growth, especially in the young stands, but both tree growth and foliar chemistry were available for 316 trees.

## Sample processing

For each tree, at least 10 leaves were composited for analysis, avoiding leaves with evidence of disease, herbivory or damage from buckshot. Leaves were oven-dried at 60 °C to constant mass and ground in a Wiley mill to pass a 40-mesh screen. Carbon and N concentrations were determined through combustion in a CN elemental analyzer (FlashEA 1112 analyzer, Thermo Scientific). Concentrations of P, Ca, Mg, and K were determined by dry ashing ∼0.25 g of ground sample at 470 °C in a muffle furnace and digesting on a hot plate with 5 or 10 mL of 6N $HNO_3$ (*Siccama et al., 1994*). The digests were analyzed by inductively coupled plasma optical emission spectrometry (ICP-OES; Optima 5300 DV, Perkin-Elmer). One blank, two replicates of standard reference material (NIST 1515 or arginine), and one duplicate sample were processed with each group of 30–40 samples. An in-house quality control followed by a blank was run after every 10 samples, and the machine was recalibrated if >5% drift was observed in the in-house standards. Errors were generally <5% (*Hong et al., 2021*).

## Data analysis

We tested for systematic interannual differences in pre-treatment foliar N and P concentrations because some trees were sampled more than once from 2008 to 2010. We used a linear mixed-effects model (nlme package in R; *Pinheiro et al., 2016*) in a nested analysis of variance (ANOVA) with year and species as fixed effects and plot (nested within stand) as a random effect where trees were the observational unit. The concentrations of N and P did not differ systematically by year for either pre-treatment foliar N ($p = 0.19$) or P ($p = 0.34$). Therefore, we used the average foliar nutrient concentration for each tree to calculate the average foliar nutrient concentration for each species within each plot as the pre-treatment covariate.

The average foliar nutrient concentration of N, P, Ca, Mg, or K for each species within each plot was the response variable for statistical analysis. We used a linear mixed-effects model in a nested analysis of covariance (ANCOVA) with pre-treatment foliar concentrations as a covariate. Pre-treatment foliar nutrient concentrations were significant in explaining variation in post-treatment concentrations ($p < 0.001$ for all five models). The predictor variables were N addition, P addition, species, stand age class, and all interactions of N addition, P addition, and species. Plots nested within stands and stands nested within forest sites (Bartlett, Hubbard Brook, and Jeffers Brook) were included as random effects. Forest stand was the unit of replication ($n = 10$). This factorial approach compared response variables in plots with N addition (*i.e.*, N and N+P plots) to those with no N addition (*i.e.*, control and P plots); similarly, plots with P addition were compared to those with no P addition. The interaction of N and P compares the response in N+P plots to that predicted by the sum of the main effects of N and P.

We also tested the relationship between pre-treatment foliar chemistry and the growth response of trees to N and P fertilization. Relative basal area growth of each tree from 2011

**Table 2  ANCOVA table of mixed effects models showing the main effects of nutrient addition and stand age class on foliar concentrations of N, P, Ca, Mg, and K, using pre-treatment foliar concentrations as covariates.** Numerator and denominator degrees of freedom (df) are separated by a comma.

| Predictor | df | Foliar N | | Foliar P | | Foliar Ca | | Foliar Mg | | Foliar K | |
|---|---|---|---|---|---|---|---|---|---|---|---|
| | | **F** | **P** | **F** | **P** | **F** | **P** | **F** | **P** | **F** | **P** |
| Pre-treatment (covariate) | 1,93 | 256.85 | **<0.001** | 229.94 | **<0.001** | 112.19 | **<0.001** | 367.45 | **<0.001** | 239.88 | **<0.001** |
| N | 1,27 | 50.73 | **<0.001** | 12.08 | **0.002** | 13.97 | **<0.001** | 8.35 | **0.008** | 0.32 | 0.58 |
| P | 1,27 | 4.24 | **0.05** | 275.20 | **<0.001** | 0.91 | 0.35 | 0.01 | 0.93 | 0.02 | 0.88 |
| Species | 5,93 | 11.08 | **<0.001** | 12.62 | **<0.001** | 3.26 | **<0.01** | 9.14 | **<0.001** | 3.80 | **0.004** |
| Age class | 2,5 | 0.99 | 0.43 | 1.56 | 0.30 | 0.07 | 0.93 | 1.68 | 0.28 | 0.78 | 0.51 |
| N × P | 1,27 | 0.18 | 0.68 | 3.24 | **0.08** | 1.21 | 0.28 | 0.004 | 0.95 | 0.03 | 0.87 |
| N × Species | 5,93 | 1.25 | 0.29 | 3.04 | **0.01** | 2.07 | **0.08** | 3.78 | **0.004** | 1.21 | 0.31 |
| P × Species | 5,93 | 0.49 | 0.79 | 16.18 | **<0.001** | 0.33 | 0.90 | 0.04 | 0.99 | 0.64 | 0.67 |
| N × P × Species | 5,93 | 0.27 | 0.93 | 2.73 | **0.02** | 0.28 | 0.92 | 0.34 | 0.88 | 0.25 | 0.94 |

**Notes.**

$p$ values < 0.05 are highlighted in boldface type.

to 2015 was the response variable, and six models tested foliar N:P and foliar concentrations of N, P, Ca, Mg, or K as predictor variables. For each linear mixed-effects model, the main effects were N addition, P addition, and one of the six foliar attributes. There were three two-way interactions and one three-way interaction. Random effects included species nested within plots, plots nested within stands, and stands nested within sites.

We used post-hoc Tukey comparisons of least-squares means to test the differences among our treatments for cases in which the main effects were significant. All statistical tests were performed using R (Version 4.1.2).

## RESULTS

### Foliar nitrogen

As expected, N addition increased foliar N concentrations, by 12% on average across ten stands and six species ($p < 0.001$ for the main effect of N addition in ANCOVA) (Tables 2 and 3; Fig. 1). More interesting was the finding that foliar N concentrations were 3% lower with P addition ($p = 0.05$ for the main effect of P addition) (Tables 2 and 3; Fig. 1). The effects of combined N and P addition on foliar N were consistent with the predicted main effects of N and P addition ($p = 0.68$ for N × P interaction).

Foliar N concentrations varied by species ($p < 0.001$ for the main effect of species in ANCOVA) but not by stand age class ($p = 0.43$) (Table 2; Fig. 1). The species with the highest foliar N was pin cherry across all treatments, which on average had foliar N 22% higher than red maple–the species with the lowest foliar N (Table 3). The response of foliar N to nutrient additions was consistent across species ($p = 0.29$ for N × species interaction and $p = 0.79$ for P × species interaction) (Table 2; Fig. 1).

### Foliar phosphorus

Foliar P concentrations across all ten stands and six species were 45% higher with P addition ($p < 0.001$ for the main effect of P addition in ANCOVA). In a manner similar to the foliar

**Table 3** Post-treatment foliar concentrations of N, P, Ca, Mg, and K by species (mean ± SE across stands).

| Treatment | N (mg/g) | P (mg/g) | Ca (mg/g) | Mg (mg/g) | K (mg/g) |
|---|---|---|---|---|---|
| *All species* | | | | | |
| CONTROL | 24.8 ± 0.6 | 1.17 ± 0.03 | 7.07 ± 0.76 | 1.65 ± 0.14 | 9.3 ± 0.3 |
| N | 27.7 ± 0.6 | 1.14 ± 0.03 | 6.53 ± 0.76 | 1.56 ± 0.14 | 9.1 ± 0.3 |
| P | 23.9 ± 0.6 | 1.75 ± 0.03 | 7.46 ± 0.76 | 1.65 ± 0.14 | 9.2 ± 0.4 |
| N+P | 26.9 ± 0.6 | 1.59 ± 0.03 | 6.53 ± 0.76 | 1.58 ± 0.14 | 9.1 ± 0.3 |
| *American beech* | *25.9 ± 0.5* | *1.24 ± 0.04* | *6.28 ± 0.75* | *1.53 ± 0.14* | *8.8 ± 0.3* |
| CONTROL | 25.1 ± 0.8 | 1.15 ± 0.06 | 6.32 ± 0.84 | 1.53 ± 0.15 | 9.0 ± 0.6 |
| N | 27.2 ± 0.8 | 1.09 ± 0.06 | 6.13 ± 0.84 | 1.51 ± 0.15 | 8.5 ± 0.6 |
| P | 24.3 ± 0.8 | 1.36 ± 0.06 | 6.64 ± 0.84 | 1.55 ± 0.15 | 8.7 ± 0.6 |
| N+P | 27.1 ± 0.8 | 1.37 ± 0.06 | 6.02 ± 0.84 | 1.52 ± 0.15 | 9.1 ± 0.6 |
| *Pin cherry* | *29.5 ± 0.8* | *1.79 ± 0.06* | *6.75 ± 0.81* | *1.97 ± 0.16* | *10.7 ± 0.6* |
| CONTROL | 28.1 ± 1.2 | 1.26 ± 0.10 | 7.24 ± 0.98 | 2.02 ± 0.19 | 11.2 ± 0.8 |
| N | 32.4 ± 1.2 | 1.20 ± 0.10 | 6.28 ± 0.98 | 1.89 ± 0.19 | 10.6 ± 0.8 |
| P | 25.7 ± 1.2 | 2.67 ± 0.10 | 7.92 ± 0.97 | 2.00 ± 0.19 | 10.7 ± 0.9 |
| N+P | 31.8 ± 1.2 | 2.03 ± 0.10 | 5.56 ± 0.97 | 1.98 ± 0.19 | 10.2 ± 0.8 |
| *Red maple* | *24.1 ± 1.0* | *1.25 ± 0.06* | *7.16 ± 0.84* | *1.46 ± 0.15* | *8.6 ± 0.6* |
| CONTROL | 23.2 ± 1.4 | 1.08 ± 0.10 | 6.81 ± 1.01 | 1.35 ± 0.19 | 8.5 ± 0.9 |
| N | 26.8 ± 1.4 | 1.15 ± 0.10 | 7.64 ± 1.02 | 1.58 ± 0.19 | 9.0 ± 0.9 |
| P | 22.2 ± 1.8 | 1.38 ± 0.14 | 6.70 ± 1.24 | 1.40 ± 0.24 | 7.9 ± 1.2 |
| N+P | 24.2 ± 1.4 | 1.38 ± 0.10 | 7.50 ± 1.01 | 1.53 ± 0.19 | 9.1 ± 0.9 |
| *Sugar maple* | *26.0 ± 0.9* | *1.34 ± 0.05* | *6.82 ± 0.80* | *1.21 ± 0.15* | *8.1 ± 0.5* |
| CONTROL | 24.4 ± 1.2 | 1.13 ± 0.10 | 7.20 ± 0.96 | 1.27 ± 0.18 | 8.6 ± 0.8 |
| N | 27.4 ± 1.3 | 1.10 ± 0.09 | 6.06 ± 0.96 | 1.19 ± 0.18 | 8.2 ± 0.8 |
| P | 24.8 ± 1.3 | 1.67 ± 0.09 | 7.29 ± 0.96 | 1.21 ± 0.18 | 8.1 ± 0.8 |
| N+P | 27.3 ± 1.3 | 1.45 ± 0.09 | 6.74 ± 0.96 | 1.18 ± 0.18 | 7.3 ± 0.8 |
| *White birch* | *24.3 ± 0.6* | *1.31 ± 0.04* | *6.81 ± 0.78* | *1.53 ± 0.14* | *8.9 ± 0.4* |
| CONTROL | 23.3 ± 1.1 | 1.13 ± 0.08 | 6.64 ± 0.92 | 1.62 ± 0.17 | 8.4 ± 0.7 |
| N | 26.0 ± 1.1 | 1.09 ± 0.08 | 6.19 ± 0.92 | 1.42 ± 0.17 | 9.2 ± 0.7 |
| P | 22.6 ± 1.1 | 1.46 ± 0.08 | 7.64 ± 0.92 | 1.56 ± 0.17 | 8.7 ± 0.7 |
| N+P | 25.1 ± 1.1 | 1.59 ± 0.08 | 6.77 ± 0.93 | 1.53 ± 0.17 | 9.4 ± 0.7 |
| *Yellow birch* | *25.2 ± 0.6* | *1.52 ± 0.04* | *7.56 ± 0.76* | *1.96 ± 0.14* | *9.9 ± 0.3* |
| CONTROL | 25.0 ± 0.9 | 1.26 ± 0.07 | 8.20 ± 0.86 | 2.14 ± 0.16 | 9.9 ± 0.6 |
| N | 26.5 ± 0.9 | 1.19 ± 0.07 | 6.86 ± 0.86 | 1.78 ± 0.16 | 9.2 ± 0.6 |
| P | 23.5 ± 0.9 | 1.94 ± 0.07 | 8.56 ± 0.87 | 2.22 ± 0.16 | 10.9 ± 0.6 |
| N+P | 25.8 ± 0.9 | 1.69 ± 0.07 | 6.60 ± 0.86 | 1.71 ± 0.16 | 9.6 ± 0.6 |

**Notes.**
Species means across treatments are highlighted in boldface type and italicized.

N response to P addition, foliar P was 7% lower when N was added ($p = 0.002$) (Tables 2 and 3; Fig. 2). In addition, foliar P was 2% lower when N and P were added together compared to the sum of main effects of N and P addition ($p = 0.08$ for N × P interaction).

Foliar P concentrations varied by species ($p < 0.001$ for the main effect of species in ANCOVA) but not by stand age class ($p = 0.30$) (Table 2; Fig. 2). Pin cherry had the highest

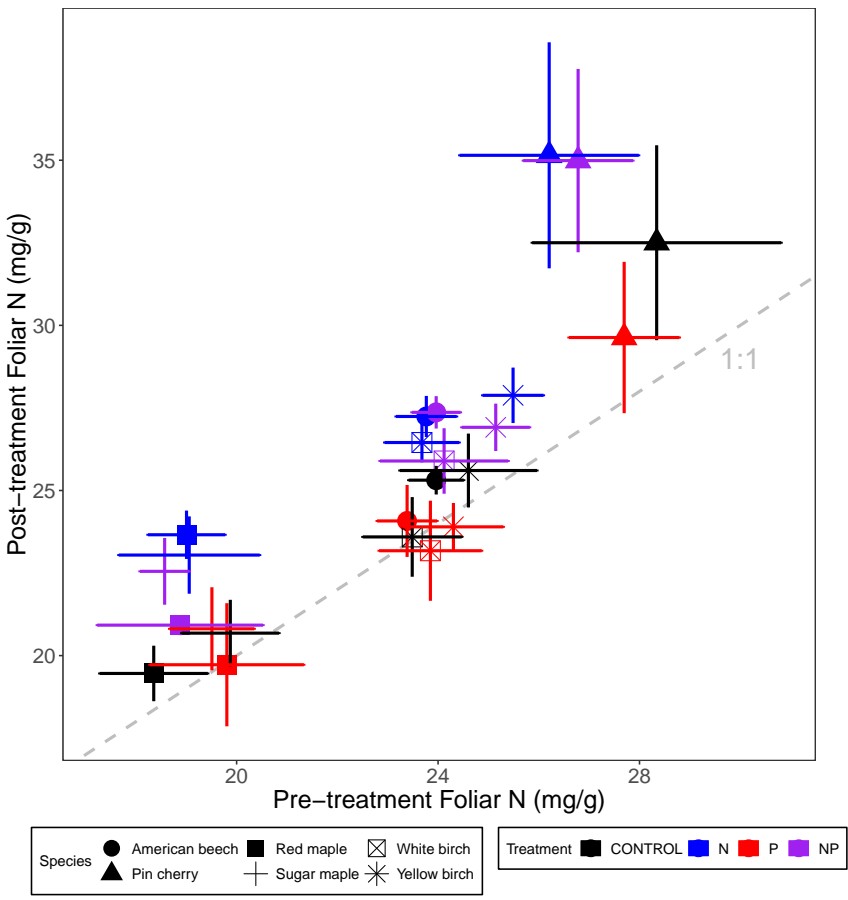

**Figure 1** **Comparison of pre-treatment (*x*-axis) and post-treatment (*y*-axis) foliar N concentrations of American beech, pin cherry, red maple, sugar maple, white birch, and yellow birch.** Points represent means across stands by treatment and error bars indicate standard errors of the mean. The dashed gray line indicates 1:1, hence increases in foliar N lie above the line and decreases fall below the line.

foliar P across all treatments, which was 44% higher, on average, than the species with the lowest foliar P–American beech (Table 3).

All the species exhibited increased P with P addition, but the greatest increase was observed for pin cherry (91%, Table 3). Foliar P of the different species exhibited distinctive responses to N addition ($p = 0.01$ for N × species interaction) and P addition ($p < 0.001$ for P × species interaction) (Table 2; Fig. 2). In particular, with N addition, foliar P was reduced for American beech, sugar maple, yellow birch, and pin cherry, with pin cherry showing the greatest reduction of 18%; in contrast, foliar P increased with N addition for red maple and white birch.

Foliar P of pin cherry differed from the other species in response to N and P added together ($p = 0.02$ for the three-way interaction of species, N, and P) (Table 2; Fig. 2). Whereas the foliar P of other species under N+P addition was within 12% of that predicted by the main effects of N and P, pin cherry had 22% lower foliar P with N+P addition

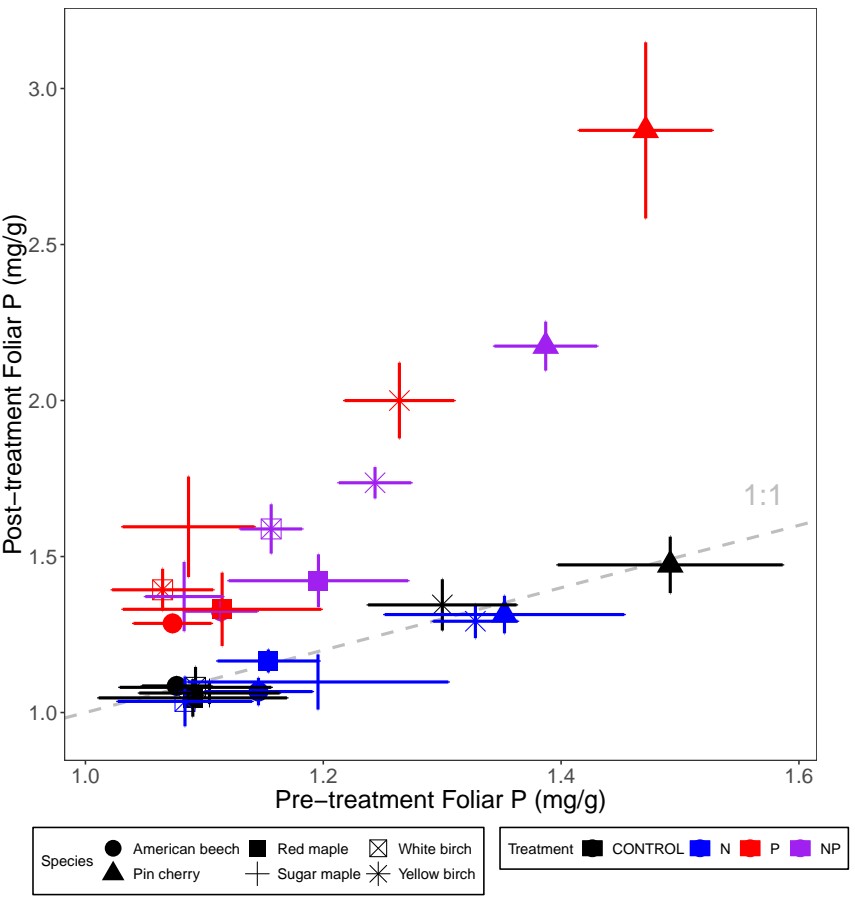

**Figure 2  Comparison of pre-treatment (*x*-axis) and post-treatment (*y*-axis) foliar P concentrations of American beech, pin cherry, red maple, sugar maple, white birch, and yellow birch.** Points represent means across stands by treatment and error bars indicate standard errors of the mean. The dashed gray line indicates 1:1, hence increases in foliar P lie above the line and decreases fall below the line.

than predicted by the sum of main effects. Without pin cherry in the model, the three-way interaction of species, N, and P was not significant ($p = 0.26$).

### Foliar Ca, Mg, and K

Foliar Ca and Mg concentrations across all six species were reduced by N addition, 10% in the case of Ca ($p < 0.001$ for the main effect of N addition in ANCOVA) and 5% for Mg ($p = 0.01$). Neither Ca nor Mg concentrations were affected by P addition ($p = 0.35$ and $p = 0.93$, respectively) (Tables 2 and 3; Figs. 3 and 4). The effects of combined N and P addition were consistent with the predicted main effects of N and P addition for foliar Ca ($p = 0.28$ for N × P interaction) and foliar Mg ($p = 0.95$).

Foliar Ca and Mg concentrations varied by species ($p < 0.01$ and $p < 0.001$, respectively, for the main effect of species in ANCOVA), but not by stand age class ($p = 0.93$ and $p = 0.28$, respectively) (Table 2; Figs. 3 and 4). The species with the highest foliar Ca was yellow birch, which on average had foliar Ca 20% higher than the lowest–American beech

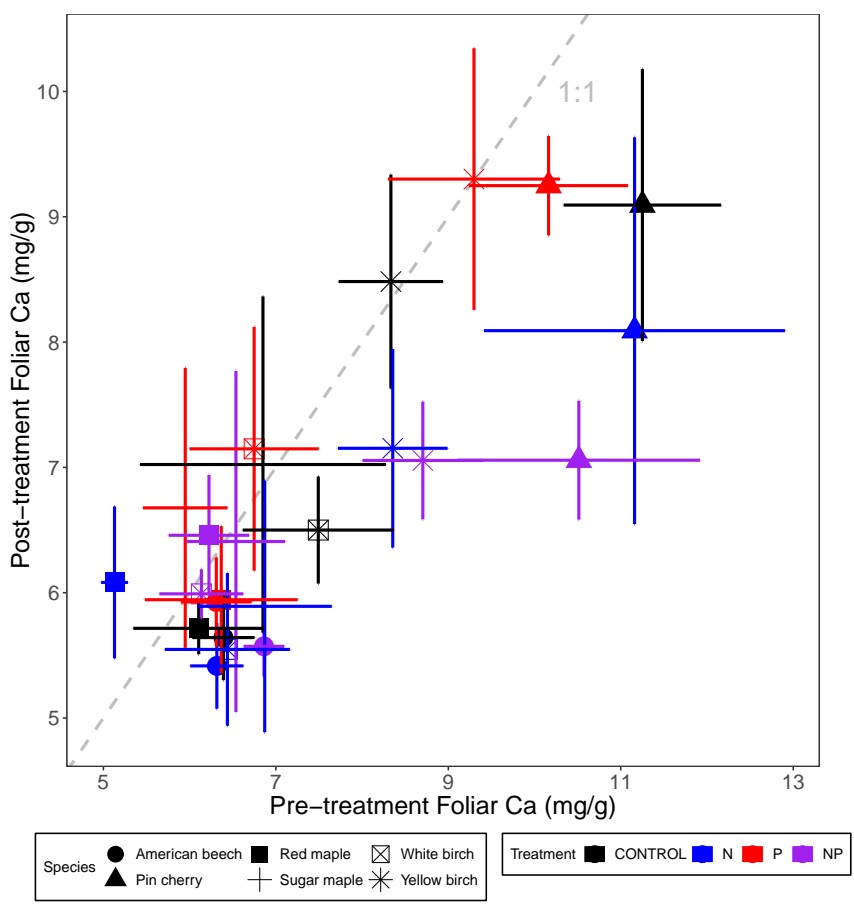

**Figure 3 Comparison of pre-treatment (*x*-axis) and post-treatment (*y*-axis) foliar Ca concentrations of American beech, pin cherry, red maple, sugar maple, white birch, and yellow birch.** Points represent means across stands by treatment and error bars indicate standard errors of the mean. The dashed gray line indicates 1:1, hence increases in foliar Ca lie above the line and decreases fall below the line.

(Table 3). The species with the highest foliar Mg was pin cherry, which on average had foliar Mg 63% higher than the lowest–sugar maple (Table 3).

Species responded differently to N addition for both foliar Ca ($p = 0.08$) and foliar Mg ($p = 0.004$ for N × species interaction), but not to P addition for either Ca ($p = 0.90$) or Mg ($p = 0.99$ for P × species interaction). Nitrogen addition reduced foliar Ca for all species except red maple, for which N addition increased foliar Ca by 12%; pin cherry showed the largest decrease of 22%. Foliar Mg, similarly, was lower with N addition for all but red maple, for which N addition increased foliar Mg by 13%; yellow birch showed the largest decrease of 20% (Table 3). There was no significant three-way interaction of species, N, and P for foliar Ca ($p = 0.92$) or foliar Mg ($p = 0.88$) (Table 2; Figs. 3 and 4).

Foliar K concentrations were not affected by nutrient additions or stand age class, but they varied by species ($p = 0.004$ for the main effect of species in ANCOVA) (Tables 2 and 3; Fig. 5). Pin cherry had the highest foliar K across all treatments, which was on average 32% higher than sugar maple–the species with the lowest foliar K (Table 3).

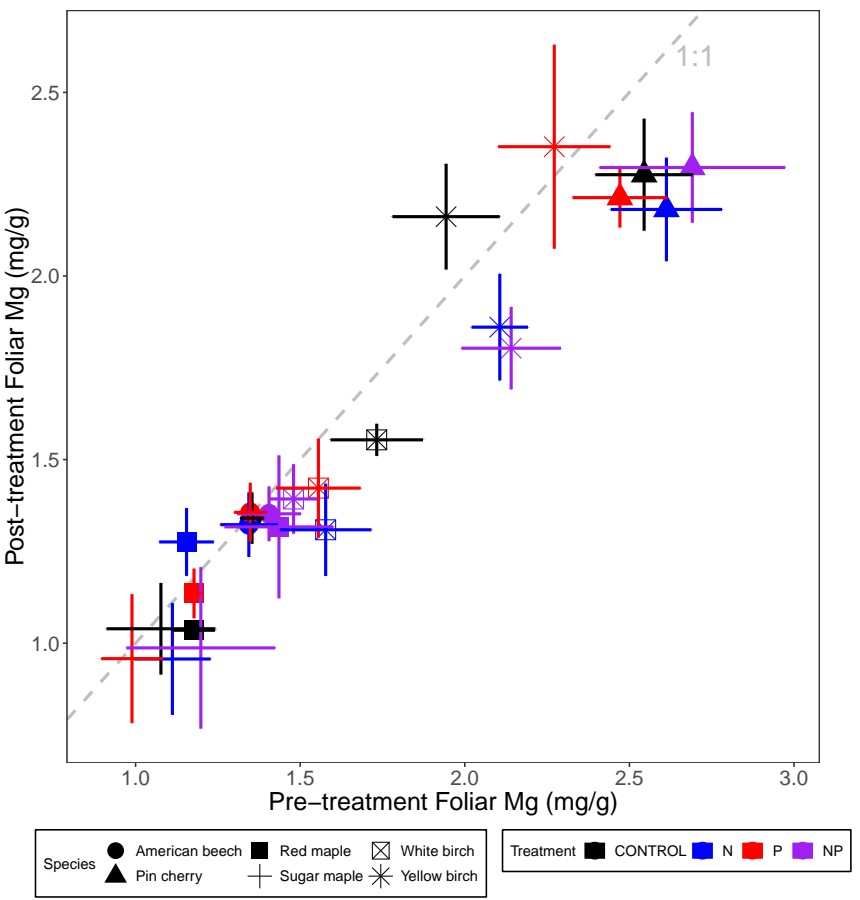

**Figure 4** **Comparison of pre-treatment (*x*-axis) and post-treatment (*y*-axis) foliar Mg concentrations of American beech, pin cherry, red maple, sugar maple, white birch, and yellow birch.** Points represent means across stands by treatment and error bars indicate standard errors of the mean. The dashed gray line indicates 1:1, hence increases in foliar Mg lie above the line and decreases fall below the line.

## Relating tree growth to pre-treatment foliar chemistry

Tree growth (relative basal area increment) from 2011 to 2015 was higher in trees with higher pre-treatment foliar N:P ($p = 0.02$ for the main effect of pre-treatment foliar N:P in ANOVA) (Table 4; Fig. 6A). This pattern was driven primarily by the fact that trees with lower foliar P had higher growth rates ($p = 0.04$) (Table 4; Fig. 6C). Other pre-treatment foliar nutrient concentrations were not significant in explaining individual tree growth rates ($p = 0.92$ for N, $p = 0.97$ for Ca, $p = 0.74$ for Mg, $p = 0.09$ for K) (Table 4; Fig. 6).

Trees in N-amended plots grew more than those in plots without N ($p < 0.02$ for the main effect of N addition) (Table 4), whereas P addition had no effect on growth ($p > 0.48$). We hypothesized that foliar N:P would explain the magnitude of growth response to N or P addition, evidenced by a statistical interaction of foliar N:P with N addition (high blue line in Fig. 6A where N:P is low) and of N:P with P addition (high red line where N:P is high). However, neither foliar N:P nor other foliar chemical attributes explained

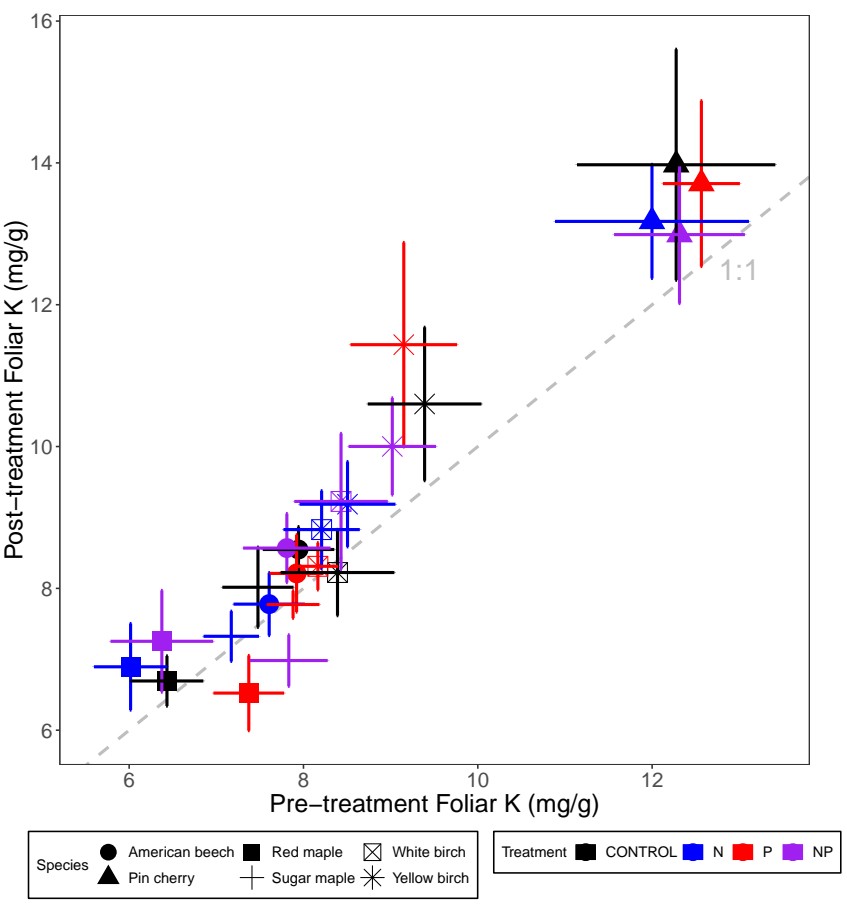

**Figure 5  Comparison of pre-treatment (*x*-axis) and post-treatment (*y*-axis) foliar K concentrations of American beech, pin cherry, red maple, sugar maple, white birch, and yellow birch.** Points represent means across stands by treatment and error bars indicate standard errors of the mean. The dashed gray line indicates 1:1, hence increases in foliar K lie above the line and decreases fall below the line.

**Table 4  ANOVA table of mixed effects models showing the main effects of foliar attribute (N:P, N, P, Ca, Mg or K) and nutrient addition on relative basal area increment from 2011 to 2015.** Numerator and denominator degrees of freedom are reported and separated by a comma.

| | | w/foliar N:P | | w/foliar N | | w/foliar P | | w/foliar Ca | | w/foliar Mg | | w/foliar K | |
|---|---|---|---|---|---|---|---|---|---|---|---|---|---|
| Predictor | df | F | P | F | P | F | P | F | P | F | P | F | P |
| Foliar attribute | 1,208 | 5.19 | **0.02** | 0.01 | 0.92 | 4.50 | **0.04** | 0.00 | 0.97 | 0.11 | 0.74 | 2.85 | 0.09 |
| N | 1,31 | 7.00 | **0.01** | 6.48 | **0.02** | 6.73 | **0.01** | 6.51 | **0.02** | 6.41 | **0.02** | 6.17 | **0.02** |
| P | 1,31 | 0.22 | 0.64 | 0.29 | 0.60 | 0.28 | 0.60 | 0.29 | 0.59 | 0.32 | 0.58 | 0.51 | 0.48 |
| Foliar Index × N | 1,208 | 0.02 | 0.89 | 0.60 | 0.44 | 0.11 | 0.74 | 0.14 | 0.71 | 0.65 | 0.42 | 1.24 | 0.27 |
| Foliar Index × P | 1,208 | 1.30 | 0.26 | 0.26 | 0.61 | 0.03 | 0.87 | 0.92 | 0.34 | 0.05 | 0.82 | 0.14 | 0.71 |
| N × P | 1,31 | 0.05 | 0.83 | 0.10 | 0.76 | 0.00 | 0.97 | 0.14 | 0.71 | 0.12 | 0.73 | 0.01 | 0.91 |
| Foliar Index × N × P | 1,208 | 0.56 | 0.45 | 0.04 | 0.84 | 0.21 | 0.65 | 2.75 | 0.10 | 0.18 | 0.68 | 0.01 | 0.94 |

**Notes.**

$p$ values < 0.05 are highlighted in boldface type.

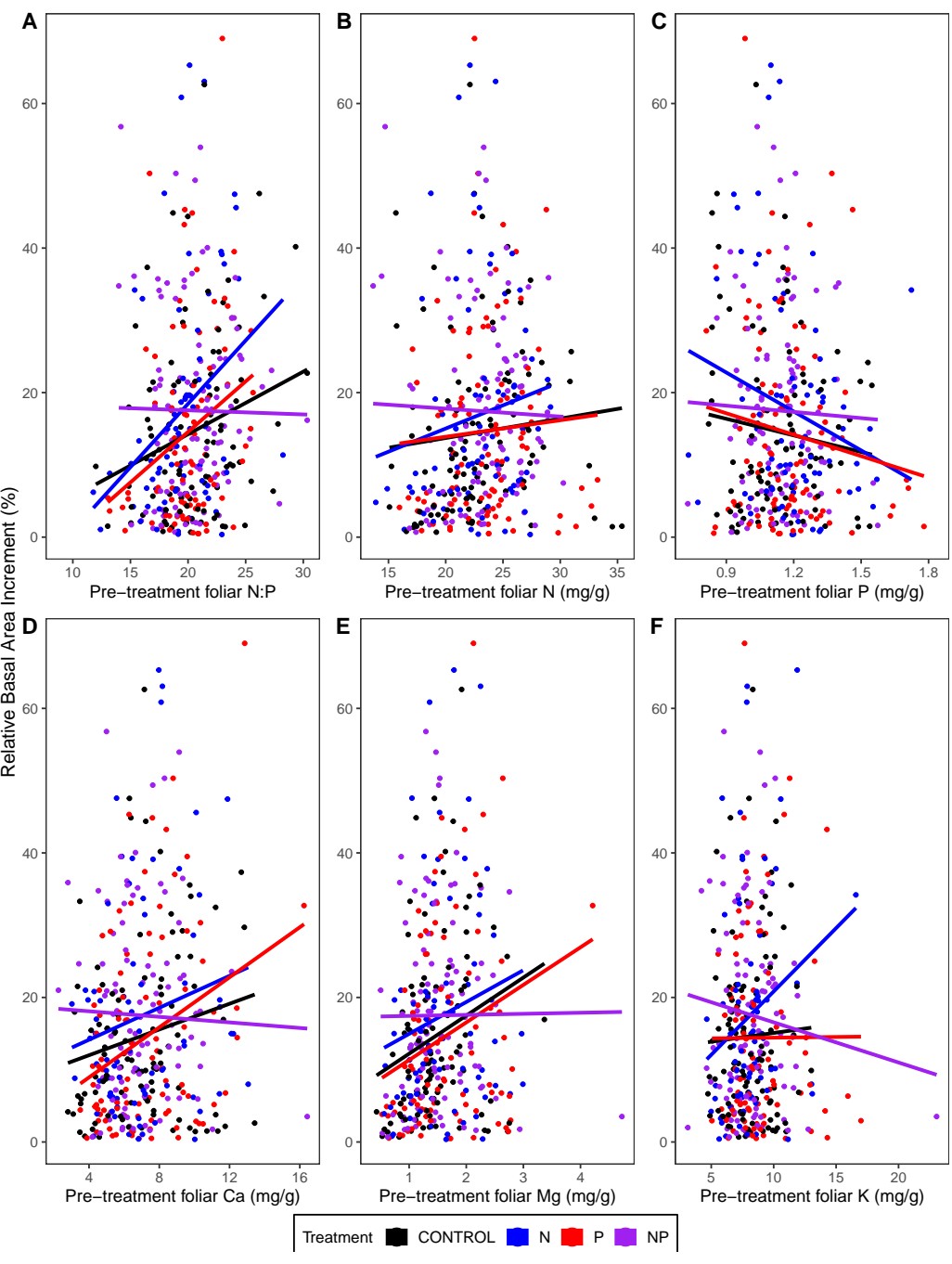

**Figure 6** **Relative basal area increments from 2011 to 2015 as a function of pre-treatment foliar (A) N:P, (B) N concentrations, (C) P concentrations, (D) Ca concentrations, (E) Mg concentrations, and (F) K concentrations.** Points represent 316 individual trees that were measured for both growth and foliar chemistry. The lines represent the linear model fit by treatment.

the growth response to N addition ($p \geq 0.27$ for interactions of N addition with foliar attributes) or P addition ($p \geq 0.26$) (Table 4; Fig. 6).

## DISCUSSION

### Nutrient limitation of tree growth

Growth of temperate forests has been thought to be primarily N-limited (*McGroddy, Daufresne & Hedin, 2004*; *Reich & Oleksyn, 2004*), but decades of anthropogenic N deposition in the northeastern United States could have altered biogeochemical cycling (*Vitousek et al., 1997*; *Stevens, David & Storkey, 2018*), possibly resulting in transactional P limitation (*Vitousek et al., 2010*). We evaluated the effectiveness of foliar nutrient analysis to distinguish between N and P limitation in a long-term factorial N × P addition experiment in northern hardwood forests (MELNHE). Foliar N:P ratios have been used to indicate the relative limitation of plant growth by N *vs.* P (*Koerselman & Meuleman, 1996*). Foliar N:P prior to treatment in the MELNHE sites (averaging across all stands and species) was 20, with red maple showing the lowest values (16) and American beech the highest (22) (Appendix B; Fig. 6). These values are high in comparison to many other forests and would predict P limitation, based on thresholds reported in the literature for other forest and vegetation types (*Herbert & Fownes, 1995*; *Ljungström & Nihlgård, 1995*; *Tessier & Raynal, 2003*). Consistent with this prediction, the average diameter growth of trees in mid-aged and mature stands between 2011 and 2015 responded more to P than N addition (*Goswami et al., 2018*).

We were surprised, therefore, to find that our trees grew more in response to N addition than P addition (Table 4; Fig. 6). The reason for this apparent discrepancy is that small trees dominated the tree inventory analyzed by *Goswami et al. (2018)* (averaging 18 cm, pre-treatment, with an interquartile range of 12–20 cm), whereas the subset of trees that we sampled were larger (averaging 24 cm, interquartile range of 14 to 31), because we sampled sun-exposed foliage from dominant and co-dominant trees. For the trees we measured, those with higher foliar N:P before treatment grew more from 2011 to 2015, associated with differences in foliar P; *i.e.*, trees with higher growth rates had lower foliar P (Table 4; Fig. 6).

Although there was considerable pre-treatment variation in foliar N:P, this variable did not explain individual tree growth responses to N or P addition (statistical interactions of N:P with nutrient addition were not significant); therefore, we could not identify a threshold diagnostic of nutrient limitation for these forests. Foliar N:P may be a less sensitive indicator of nutrient limitation of growth of large trees than previously thought, perhaps in part because trees have adaptive mechanisms to store nutrients in perennial tissues to buffer short-term deficits in soil nutrient availability. For example, in a large-scale nutrient fertilization in the Peruvian Andes and adjacent lowland Amazonia, adding N and/or P enhanced tree diameter growth and microbial and root respiration, with no detectable change in foliar chemistry (*Fisher et al., 2013*). Consistent with this hypothesis is the observation that fine root growth in three of the mature MELNHE stands was higher in N-amended than unamended plots (*Shan et al., 2022*).

## Nutrient addition effects on foliar nutrient concentrations

Addition of N or P generally resulted in increased foliar concentrations of that nutrient across tree species and stand age classes (Table 3; Figs. 1 and 2), as demonstrated in many other forest fertilization studies (*Güsewell, Koerselman & Verhoeven, 2003*; *Niinemets & Kull, 2005*; *Lu et al., 2013*). Additions of N and P have been shown to increase soil availability of the respective nutrient in our sites (*Fisk et al., 2014*), and high foliar nutrient concentrations typically reflect increased uptake of available soil nutrients (*Mugasha, Pluth & Macdonald, 1999*; *Sariyildiz & Anderson, 2005*).

The relative effects of P addition on foliar P were much greater than those of N addition on foliar N: foliar P increased by 45% compared to an average increase of 12% in foliar N, across stands and species (Figs. 1 and 2). Our rate of P addition was high relative to N, with an N:P of 3, much lower than the ratios of these nutrients in foliage. This high rate of P addition was chosen because P, but not N, is strongly and often irreversibly occluded in acid soils such as those in our study area (*Wood, Bormann & Voight, 1984*; *Cross & Schlesinger, 1995*). In fact, the relative effect of N addition on resin-available N in the organic soil horizon in the MELNHE study was twice as great as that of P addition on resin-available P (*Goswami et al., 2018*). Thus, the greater response in foliar P to P addition may reflect luxury uptake (*Van Wijk et al., 2003*). From 2011 and 2015, adding N increased the growth rate of trees, suggesting N limitation, but added P accumulated in the foliage, and P addition did not increase the diameter growth of the trees we studied.

Overall, foliar N and P concentrations were slightly but significantly suppressed by the addition of the other nutrient (Table 3; Figs. 1 and 2). Across tropical and temperate forests, grasslands, and wetlands, P concentrations in leaves and other aboveground tissues (but not roots) decreased under N addition (*Deng et al., 2017*). One explanation for such reductions in nutrient concentrations could be a dilution effect as the addition of a limiting nutrient leads to enhanced growth. Where N addition stimulates primary production, tissue P concentrations usually are reduced (*Phoenix et al., 2003*; *Menge & Field, 2007*; *Perring et al., 2008*; *Ostertag, 2010*), as demonstrated in a meta-analysis spanning multiple ecosystem types and environmental conditions (*Mao et al., 2020*).

Although adding P had no effect on foliar base cations, adding N decreased foliar Ca and Mg (Table 3; Figs. 3 and 4). Nitrogen addition reduced foliar Ca and Mg across multiple biomes, especially in woody plants and in forest ecosystems (*Mao et al., 2020*). These concomitant decreases in foliar Ca and Mg concentrations might result from dilution effects, consistent with the growth response to N addition (Table 4; Fig. 6). Another possible explanation for the decreases in foliar Ca and Mg under N addition is decreased soil $Ca^{2+}$ and $Mg^{2+}$ availability (*Lucas et al., 2011*; *Lu et al., 2014*), resulting from N-induced loss of these soil base cations via leaching leading to soil acidification (*Tian & Niu, 2015*). Between 2009 and 2017, the pH of the upper mineral soil layer decreased by 0.2 under N addition across our stands in Bartlett and Hubbard Brook (*Fisk, 2022*). However, we have not detected a direct effect of soil pH or nutrient addition on soil extractable Ca or Mg concentrations (*Walsh, 2022*).

## Species responses to nutrient addition

In contrast to the average responses across all species, foliar P concentrations of red maple and white birch increased significantly with N addition. Addition of N can enhance acquisition of P by plants via stimulation of phosphatase activity in plant roots and soil (*Olander & Vitousek, 2000*; *Phoenix et al., 2004*; *Marklein & Houlton, 2012*). For white birch, this increase was mainly associated with high foliar P in N+P plots; foliar P was actually slightly lower when only N was added (Table 3; Fig. 2). Northern hardwood tree species are known to differ in cycling and retention of N (*Lovett et al., 2004*) and perhaps in their ability to use added N for phosphatase production. Species are also known to have different strategies (*e.g.*, efficient P uptake or high P resorption) in P-limited sites (*Fujita et al., 2010*). In species associated with arbuscular mycorrhizal fungi, N addition enhances hyphal production, improving P uptake (*Schultz, 1982*; *Phillips, Brzostek & Midgley, 2013*). However, in our study, none of the responses we measured differed systematically between species with these two contrasting types of mycorrhizal associations: only red maple demonstrated higher P acquisition under N addition, whereas the other two arbuscular-mycorrhizal species–pin cherry and sugar maple–did not.

Foliar nutrient concentrations often differ markedly among tree species in the same forest, and their responses to nutrient additions may be associated in part with differences in life-history strategy (*Goddard & Hollis, 1984*; *Binkley, 1986*). In our study, the extremely shade-intolerant species, pin cherry (*Marks, 1974*), exhibited the highest foliar nutrient concentrations pre-treatment and also the largest responses to nutrient additions. These results were consistent with an earlier study conducted in the same forest type where N and P enrichment of foliage in response to fertilization were more pronounced for pin cherry than for maples, beech or birches (*Fahey, Battles & Wilson, 1998*). Pin cherry is a fast-growing, short-lived species that dominates early stages of secondary succession when resources are abundant (*Auchmoody, 1979*; *Bormann & Likens, 1979*), and this exploitative strategy allows pin cherry to effectively acquire soil nutrients, including luxury uptake (*Mou, Fahey & Hughes, 1993*). The less pronounced differences in life history among the other five tree species in these stands did not produce detectable differences in foliar nutrient responses to changes in N and P availability. Whether this observation can be generalized across other mixed forests awaits further investigation.

## CONCLUSION

Although foliar N:P ratios have been regarded as an effective diagnostic tool to distinguish N *vs.* P limitation of productivity in terrestrial vegetation, we were unable to demonstrate such a relationship in our long-term factorial N × P addition experiment in northern hardwood forests. Although high pre-treatment foliar N:P ratios (mean = 20) suggested P limitation, we observed increased individual tree growth in response to N but not P addition. Foliar nutrient concentrations responded as expected to nutrient additions, and significant decreases in concentrations of one nutrient were associated with the addition of the other nutrient; whether these decreases influence tree performance warrants additional study. Across the six tree species studied, the only significant species effects on foliar nutrient

responses were associated with the extreme exploitive species, pin cherry, which showed by far the greatest foliar N and P responses to nutrient addition; apparently differences in life history characteristics among the other five dominant species (*e.g.*, mycorrhizal type, shade tolerance) did not explain foliar nutrient responses to N and P addition.

## ACKNOWLEDGEMENTS

Matt Vadeboncoeur, Steve Hamburg, Mary Arthur, Marty Acker, and Melany Fisk helped to establish the MELNHE plots from 2004 to 2010. Maintaining and fertilizing the plots has involved 38 graduate students and >80 field crew members. R. Quinn Thomas, Issac Lombard, Corrie Bloddgett, Bali Quintero, Craig See, and William O'Neill (pre-treatment) and Adam Wild, Gretchen Dillon, Kara Gonzales, and Daniel Hong (post-treatment) helped to collect and analyze samples. Chuck Schirmer, Deb Driscoll, Marlene Braun, and Jeff Merriam were key to sample analysis, and Mary Hagemann and Christine Costello made our lab and field operations run smoothly. The MELNHE project (http://www.esf.edu/melnhe) contributes to the Hubbard Brook Ecosystem Study. The Hubbard Brook and Bartlett Experimental Forests are operated by the USDA Forest Service's Northern Research Station.

### Funding

This work was supported by the USDA National Institute of Food and Agriculture (2019-67019-29464) and the National Science Foundation (DEB-0949324) including the Long-Term Ecological Research Program (DEB-0423259, DEB-1114804, and DEB-1637685). The funders had no role in study design, data collection and analysis, decision to publish, or preparation of the manuscript.

### Grant Disclosures

The following grant information was disclosed by the authors:
USDA National Institute of Food and Agriculture: 2019-67019-29464.
National Science Foundation: DEB-0949324.
Long-Term Ecological Research Program: DEB-0423259, DEB-1114804, DEB-1637685.

### Competing Interests

The authors declare there are no competing interests.

### Author Contributions

- Daniel S. Hong conceived and designed the experiments, performed the experiments, analyzed the data, prepared figures and/or tables, authored or reviewed drafts of the paper, and approved the final draft.
- Kara E. Gonzales conceived and designed the experiments, performed experiments, analyzed the data, authored or reviewed drafts of the paper, and approved the final draft.
- Timothy J. Fahey and Ruth D. Yanai conceived and designed the experiments, authored or reviewed drafts of the paper, and approved the final draft.

## Data Availability

The data are available in the Environmental Data Initiative repository: Hong, S.D., K.E. Gonzales, C.R. See, and R.D. Yanai. 2021. MELNHE: Foliar Chemistry 2008-2016 in Bartlett, Hubbard Brook, and Jeffers Brook (12 stands) ver 1. Environmental Data Initiative https://doi.org/10.6073/pasta/b23deb8e1ccf1c1413382bf911c6be19.

## Supplemental Information

Supplemental information for this article can be found online at http://dx.doi.org/10.7717/peerj.13193#supplemental-information.

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
