# Peer review of "Foliar nutrient concentrations of six northern hardwood species responded to nitrogen and phosphorus fertilization but did not predict tree growth"

_PeerJ, doi:10.7717/peerj.13193_

## Round 0.1 · original submission · Major Revisions

Your manuscript was reviewed by two experts who suggested that your manuscript needs revision before further consideration for publication. I invite you to revise the manuscript.

Reviewer 1 ·

Basic reporting

The MELNHE experiment is a great service to forest biogeochemistry, and this paper represents an interesting update from the ongoing study. Context from the larger experiment is communicated, and these findings are unique and worthy of their own paper. However, the framing of the data requires revision before publication.

Experimental design

The data are rigorously collected and reported. I particularly appreciate the reporting of negative results (mycorrhizal type, stand age). Nonetheless, the busy figures suggest a research question that could use more honing. Why show the slope of foliar nutrient change for all treatment types combined? If a key hypothesis is that N addition will stimulate foliar P (or vice versa), panels of the main text figures should contrast treatment types and combine the factors that were not statistically interesting (like every species except pin cherry).

Validity of the findings

The work of this manuscript is framed in part as using foliar nutrient response to fertilization as an alternative measure of nutrient limitation to productivity. However, this manuscript is better viewed as a direct test of whether foliar nutrition changes CAN indicate nutrient limitation to productivity. It is akin to a methods test where a direct measure of nutrient limitation (tree trunk growth) is carried out in tandem with the controversial method of foliar chemistry. The authors are correct to consider the possibility that nutrient limitation was missed by their tree trunk measurements. An unmeasured biomass pool might have been stimulated by N+P fertilization. However, especially because tree trunks are a large share of NPP in temperate forests and are perhaps the most important pool for decadal-scale carbon modeling of forests, the authors should also consider the possibility that their findings suggest foliar nutrition is not a useful metric of limitation.

Additional comments

Thank you for your work doing this research and preparing this manuscript. My biggest piece of feedback is to reformat the figures to address a research question. Here are specific notes line by line.

Line 38
Unhealthy leaves were excluded from your study, and there are open questions about the usefulness of foliar nutrition for assessing nutrient limitation (One example: Fisher JB, Malhi Y, Torres IC, Metcalfe DB, van de Weg MJ, Meir P, Silva-Espejo JE, Huasco WH. 2013. Nutrient limitation in rainforests and cloud forests along a 3,000-m elevation gradient in the Peruvian Andes. Oecologia 172:889–902.)

Line 80-82
This idea was originally proposed in: Koerselman W, Meuleman AFM. 1996. The Vegetation N: P Ratio: a New Tool to Detect the Nature of Nutrient Limitation. J Appl Ecol 33:1441–1450.

Line 107-109
More careful distinction is needed between this “study” within the long-term experiment (foliar nutrient response), and the long-term experiment as a whole (MELNHE). This sentence also seems out of place when MELNHE is introduced later on.

Line 109-111
Broad justification of the long-term experiment is not necessary. The manuscript should focus on justifying the questions it addresses within the long-term experiment.

Line 117-121
I suggest you state research questions here and focus on answering them throughout the paper. These are two research questions I'd propose, based on what I find compelling in your data. 1. Can foliar chemistry (or changes in foliar chemistry) predict nutrient limitation to productivity, as measured by tree trunk growth? 2. Does adding one nutrient stimulate, inhibit, or not affect the status of another nutrient in foliage?

Line 158-159
Did you perform a sensitivity analysis on the inclusion of these edge trees?

Line 167-170
How many trees were sampled from edges?

Line 176-194
I like that you are sharing so many details about your protocol for future workers, but it is a bit distracting from the main story. Your main story is deserving of attention! I think this should go in the supplemental.

Line 200-201
I think I’ve missed something. Were pre-treatment values measured three times (2008,2009,2010) in every plot? Also, what about Ca, Mg, and K?

Line 229-230
Could this have been affected by your choice to designate the variables carrying stand age information as random effects?

Line 260-262
The conventional placement of P in the denominator can lead to overstating the relative role of P as compared to N. Use of log values is one solution, see: "The misuse of ratios in ecological stoichiometry" by Isles (2020) in Ecology.

Line 288-298
Vertical profiles of Ca, Mg, and K could be quite different than vertical profiles of N and P. Plants might have more reason or ability to concentrate N and P in the uppermost sunlit leaves you sampled. The tiny K+ ion, in particular, is highly susceptible to leaching in upper leaves but less so in shaded leaves because throughfall water is loaded with K by the time it reaches shaded leaves. Vertical leaf trait profiles have been published for sugar maple (e.g. Coble & Cavaleri's 2017 paper in Tree Physiology) and probably for your other species as well.

Line 315-316
By N allowing greater use of P for growth, are you suggesting N addition mobilized P to wood? roots? reproduction? If so, you might be interested in: Sardans J, Pen ̃uelas J. 2014. Trees increase their P: N ratio with size: Phosphorus; the treasure nutrient. Glob Ecol Biogeogr 24:147–156.

Line 319-320
Was the degree of P limitation (i.e. magnitude of tree trunk growth response to P addition) proportional to pre-treatment foliar N:P (supporting Koerselman & Meuleman’s 1996 hypothesis), magnitude foliar nutrient response to P application (evidence for the utility of foliar chemistry in assessing limitation to productivity), or neither (evidence against the utility of foliar chemistry in assessing limitation to productivity) ?

Line 390-392
I appreciate this context about the importance of Ca at the site. What are the roles of Mg and K? If site specific data do not exist, discussion of their role in temperate forests or even plants in general would help readers understand the motivation for the measurements.

Reviewer 2 ·

Basic reporting

This manuscript describes a unique experiment evaluating fertilizer additions of N and P and the effects on foliar nutrients on tree species in a northern hardwood forest. The paper is well prepared, the figures are relevant, and the literature used in the paper is up to date and provides excellent context.

My only questions are in the use of the term "foliar base cations" -- usually referring to Ca and Mg (and sometimes K), which while they act as base cations in the soil, in the foliage are generally without charge, and it would be better to simply say something like "foliar Ca and Mg" .

Second, the term "tolerant" is often used to mean "shade tolerant". Because plants have other tolerances (or lack thereof) it is best to be specific at each usage.

Why is N:P the only ratio evaluated? The rational/context for only N:P could be better described.

Experimental design

Very well designed, and the QA/QC description is very thorough.

Validity of the findings

I find the use of relative changes, expressed as percentages, potentially misleading. I accept that the authors do not actually statistically compare them, but a small change in a small number could result in a percentage similar to a large change in a large number. The data are available to the reader, so they can be evaluated, fortunately.

In the introduction, the objective statement includes the statement "We anticipated that tree species, stand age and site quality would also influence foliar responses." Yet there is little if any, reference to site quality in the discussion and conclusions . It would be good to revisit the hypotheses/objectives clearly.

Additional comments

Well done study and very well done manuscript.

---

## Round 0.2 · Minor Revisions

One of our reviewers has reassessed your revised manuscript and suggested making a few minor changes. I invite you to revise the manuscript.

Reviewer 1 ·

Basic reporting

My major revisions were addressed, particularly in framing a stronger research question. This is a rigorous paper and a great addition to the literature. I just have a few very small comments. I will not need to see the paper again.

Experimental design

no comment

Validity of the findings

Figure 6: Are any of these trends significant? And if so, significantly different from each other? I am particularly interested in whether N+P addition eliminated the growth advantage of leaves that had higher Ca and Mg levels at the start of the study...and how N+P treatment perhaps even made a high starting potassium value into a disadvantage in growth.

Line 138: Were the exact same cations measured in the Fang study?

Line 1322: This comment about white birch demonstrates to me why N+P and +N should be treated separately. Someone interested in whether trees can leverage added N to get more P would not be convinced by findings that include N+P data. Combining the data might be relevant for some other research questions but not the one discussed here.

---

## Round 0.3 · accepted · Accept

Thank you for addressing the remaining reviewer comments.